# Choice of CTO scores to predict procedural success in clinical practice. A comparison of 4 different CTO PCI scores in a comprehensive national registry including expert and learning CTO operators

Pablo Salinas[1]*, Nieves Gonzalo[1], Víctor H. Moreno[1], Manuel Fuentes[2], Sandra Santos-Martinez[3], José Antonio Fernandez-Diaz[4], Ignacio J. Amat-Santos[3], Francisco Bosa Ojeda[5], Juan Caballero Borrego[6], Javier Cuesta[7], José María de la Torre Hernández[8], Alejandro Diego-Nieto[9], Daniela Dubois[10], Guillermo Galeote[11], Javier Goicolea[4], Alejandro Gutiérrez[12], Miriam Jiménez-Fernández[6¤], Jesús Jiménez-Mazuecos[13], Alfonso Jurado[11,14], Javier Lacunza[15], Dae-Hyun Lee[8], María López[16], Fernando Lozano[14], Javier Martin-Moreiras[9], Victoria Martin-Yuste[17], Raúl Millán[10], Gema Miñana[18], Mohsen Mohandes[19], Francisco J. Morales-Ponce[20], Julio Núñez[18], Soledad Ojeda[21], Manuel Pan[21], Fernando Rivero[7], Javier Robles[22], Sergio Rodríguez-Leiras[23], Sergio Rojas[19], Juan Rondán[24], Eva Rumiz[25], Manel Sabaté[17], Juan Sanchís[18], Beatriz Vaquerizo[10], Javier Escaned[1]

1 Cardiology Department, Hospital Clínico San Carlos, Instituto de Investigación Sanitaria del Hospital Clínico San Carlos (IdISSC), Madrid, Spain, 2 Servicio de Medicina Preventiva, Hospital Clínico San Carlos, Instituto de Investigación Sanitaria San Carlos (IdISSC), Madrid, Spain, 3 Instituto de Ciencias del Corazón (ICICOR), Hospital Clínico Universitario de Valladolid, Valladolid, Spain, 4 Interventional Cardiology Department, Hospital Universitario Puerta de Hierro, Majadahonda, Spain, 5 Servicio de Cardiología, H. Tenerife, Tenerife, Spain, 6 Servicio de Cardiología, HU. San Cecilio, Granada, Spain, 7 Servicio de Cardiología, H. de la Princesa, Madrid, Spain, 8 Servicio de Cardiología, H. Valdecilla, Santander, Spain, 9 Servicio de Cardiología, Complejo Asistencial Universitario de Salamanca, IBSAL, CIBERCV, Salamanca, España, 10 Servicio de Cardiología, H. del Mar, Barcelona, Spain, 11 Servicio de Cardiología, H. la Paz, Madrid, Spain, 12 Servicio de Cardiología, H. Jerez, Jerez, Spain, 13 Servicio de Cardiología, H. Albacete, Albacete, Spain, 14 Servicio de Cardiología, H. Ciudad Real, Ciudad Real, Spain, 15 Servicio de Cardiología, H. de la Arrixaca, Murcia, Spain, 16 Servicio de Cardiología, H. León, León, Spain, 17 CIBER CV, IDIBAPS, Instituto Cardiovascular, Servicio de Cardiología, H. Clinic Barcelona, Spain, 18 Servicio de Cardiología, H. Clínico de Valencia. Universidad de Valencia, CIBERCV, Valencia, Spain, 19 Servicio de Cardiología, H. Joan XXIII, Tarragona, Spain, 20 Servicio de Cardiología, H. Puerto Real, Puerto Real, Spain, 21 Reina Sofia Hospital, Maimonides Institute for Research in Biomedicine of Córdoba (IMIBIC), University of Córdoba, Córdoba, Spain, 22 Servicio de Cardiología, H. Burgos, Burgos, Spain, 23 Servicio de Cardiología, H. Virgen de la Macarena, Málaga, Spain, 24 Servicio de Cardiología, H. Cabueñes, Gijón, Spain, 25 Servicio de Cardiología, H. General de Valencia, Valencia, Spain

¤ Current address: Servicio de Cardiología, HU Virgen de las Nieves, Granada, Spain
* salinas.pablo@gmail.com

## Abstract

### Background

We aimed to compare the performance of the recent CASTLE score to J-CTO, CL and PROGRESS CTO scores in a comprehensive database of percutaneous coronary intervention of chronic total occlusion procedures.

**Data Availability Statement:** The data underlying the results presented in the study are available from Spanish Association of Interventional Cardiology (Iberian CTO Registry) at https://www. hemodinamica.com/cientifico/registros-y-trabajos/ registros-y-trabajos-actuales/registro-iberico-de-oclusiones-cronicas/.

**Funding:** The author(s) received no specific funding for this work.

**Competing interests:** Abbott provided an unrestricted grant to support the Registry and this publication's fees. There are no patents, products in development or marketed products associated with this research to declare. This does not alter our adherence to PLOS ONE policies on sharing data and materials.

**Abbreviations:** AUC, Area Under the Curve; CABG, Coronary Artery Bypass Graft; CTO, Chronic Total Occlusion; IDI, Integrated Discrimination Improvement; LVEF, Left Ventricular Ejection Fraction; LAD, Left Anterior Descending; MI, Myocardial Infarction; NRI, Net Reclassification Index; PCI, Percutaneous Coronary intervention; PPV, Positive Predictive Value; ROC, Receiver-operator characteristic.

## Methods

Scores were calculated using raw data from 1,342 chronic total occlusion procedures included in REBECO Registry that includes learning and expert operators. Calibration, discrimination and reclassification were evaluated and compared.

## Results

Mean score values were: CASTLE 1.60±1.10, J-CTO 2.15±1.24, PROGRESS 1.68±0.94 and CL 2.52±1.52 points. The overall percutaneous coronary intervention success rate was 77.8%. Calibration was good for CASTLE and CL, but not for J-CTO or PROGRESS scores. Discrimination: the area under the curve (AUC) of CASTLE (0.633) was significantly higher than PROGRESS (0.557) and similar to J-CTO (0.628) and CL (0.652). Reclassification: CASTLE, as assessed by integrated discrimination improvement, was superior to PROGRESS (integrated discrimination improvement +0.036, p<0.001), similar to J-CTO and slightly inferior to CL score (– 0.011, p = 0.004). Regarding net reclassification improvement, CASTLE reclassified better than PROGRESS (overall continuous net reclassification improvement 0.379, p<0.001) in roughly 20% of cases.

## Conclusion

Procedural percutaneous coronary intervention difficulty is not consistently depicted by available chronic total occlusion scores and is influenced by the characteristics of each chronic total occlusion cohort. In our study population, including expert and learning operators, the CASTLE score had slightly better overall performance along with CL score. However, we found only intermediate performance in the c-statistic predicting chronic total occlusion success among all scores.

## Background

The percutaneous coronary intervention (PCI) of a Chronic Total Occlusion (CTO) is currently one of the most complex procedures in interventional cardiology. Compared with non-CTO PCI, interventions in chronically occluded vessels take more time, toolbox resources, radiation exposure and risk of complications [1–3]. Therefore, the patients should have a comprehensive pre-procedural assessment, including symptoms, ischemia and viability testing, in order to make a straightforward clinical indication. Currently, patients are derived for CTO PCI to improve patient's symptoms, to reduce significant ischemia burden or to seek complete revascularization to improve left ventricular ejection fraction (LVEF) [2, 4].

Once the clinical indication is established, the operator must have a realistic estimation of the probability of procedural success that will eventually be discussed with the patient and used for clinical decision-making. The procedural difficulty is largely dictated by structural characteristics of the CTO, the coronary anatomy and clinical factors. Several scores have been developed over the last ten years to integrate these variables and perform an objective assessment of procedural CTO difficulty. Following the Multicenter CTO registry in Japan (J-CTO) score [5], the Clinical and Lesion-related (CL) [6] and Prospective Global Registry for the Study of Chronic Total Occlusion Intervention (PROGRESS CTO) scores [7] were developed and tested in study populations. More recently, the CASTLE score has been proposed on the

grounds of a number of reasons [8]. First, its derivation dataset is by far the largest (14,882 patients from the EuroCTO registry compared to 329, 1143 and 521 patients in J-CTO, CL and PROGRESS scores, respectively). Second, it is representative of a large number of different European centres and operators encompassing the wholes spectrum of CTO-PCI approaches. And third, it's the most recent and thus might reflect the impact of novel contemporary devices in CTO recanalization success.

In this study we aim to compare the performance of the new CASTLE score to the previous and representative J-CTO, CL and PROGRESS CTO scores using an extensive database of CTO-PCI procedures (the Iberian Registry of CTO PCI, or REBECO). In brief, the Association of Interventional Cardiology of the Spanish Society of Cardiology prompted an open initiative to gather prospective CTO PCI data across Spain. This ongoing Registry involves centres with a variety of expertise in CTO PCI with a pragmatic character intending to record every CTO attempt in daily practice into a real-world data-set.

## Methods

The methodology, variable definitions and first results of the Iberian Registry of CTO PCI are discussed elsewhere [9]. For the present analysis (n = 1626 CTO cases), we used a data extraction from 24 centers taken August 31, 2019 (cases belong to the timeframe 2015–2019). From this database, we selected those cases with valid values on all critical variables used to estimate the four scores plus the variable CTO technical success (n = 1342 CTO cases). Technical success was defined as CTO recanalization with final TIMI 3 flow. Subsequently, the J-CTO, CL, PROGRESS and CASTLE scores were independently calculated from the raw registry data (Table 1 summarizes the score definitions of each score taken from the original publications [5–8]). Each score was dichotomized in simple or complex cases for secondary analysis using cutoffs chosen from the clinical practice [10] (CASTLE <4 vs ≥4, J-CTO <3 vs ≥3, PROG-RESS CTO <3 vs ≥3, CL-SCORE <5.5 vs ≥5.5).

Qualitative variables were summarized by frequency distribution, and quantitative variables as mean values and SDs. Continuous, non-normally distributed variables were expressed as

**Table 1. Scoring system for each of the scores used in this study.**

| SCORE CATEGORIES | CASTLE 7 (0 to 6) | J-CTO 6 (0 to 5) | PROGRESS CTO 5 (0 to 4) | CL 15 (0 to 8 by 0.5) |
|---|---|---|---|---|
| **CABG history** | CABG history (yes) | | | CABG history‡ (yes) |
| **MI history** | | | | MI history (yes) |
| **Age** | ≥70 | | | |
| **Stump** | Blunt or invisible | Blunt | Poor cap visualization or non-tapered stump | Blunt |
| **Tortuosity** | Severe (≥2 bends >90˚ or 1 bend >120˚) or unseen | 1 Bending >45˚ | Moderate or Severe (2 bends>70˚ or 1 bend>90˚) | |
| **Long lesion** | ≥20 mm (visual estimation) | ≥20 mm (visual estimation) | | ≥20 mm† (visual estimation) |
| **Calcification** | Severe (≥50% CTO segment) | Presence of any calcification | | Severe‡ (out of 3 categories) |
| **Redo** | | Yes | | |
| **Interventional collaterals** | | | Absence | |
| **CTO Location** | | | Circumflex | Non-LAD |

Each item scores 1 point except (†) 1.5 points and (‡) 2 points. Definitions as per original references [5–8]. CABG, Coronary Artery Bypass Graft. CTO, Chronic Total Occlusion. MI, Myocardial Infarction. LAD, Left Anterior Descending

medians and interquartile ranges (IQR). Chi-Square linear p for trend was estimated for observed success rates across strata of the four scores. Hosmer-Lemeshow goodness-of-fit test, obtained by univariate logistic regression with the success rate as the dependent variable, was used to assess the calibration of the scores. Discrimination was analyzed with the area under the curve (AUC) of receiver-operator characteristics (ROC) curve. Comparison of the AUC, taking CASTLE AUC as a reference, from receiver-operating characteristic curve analysis was performed with the DeLong method [11] and p-values were corrected by Bonferroni method. To assess discrimination and reclassification ability, each score was compared with the CASTLE score as a reference by absolute integrated discrimination improvement (IDI) index, as well as continuous net reclassification index (NRI) [12]. Sensitivity, specificity, positive and negative predictive values for technical success were calculated for the complex case cutoffs previously defined by expert consensus and for the highest Youden index values. Statistical analysis was performed with STATA version 15.0 and IBM SPSS Statistics Version 21.0 (IBM Corporation, Chicago, Illinois). A 2-tailed p-value of <0.05 was considered statistically significant.

## Results

Clinical and interventional characteristics of the 1,342 patients included in the study are shown in Tables 2 & 3. Mean score for CASTLE was 1.60±1.10; J-CTO 2.15±1.24; PROGRESS 1.68±0.94 and CL 2.52±1.52. The overall success rate in the patients included in the study was 77.8%. Fig 1 shows the scoring distribution among the study sample compared to each score's original derivation cohort [5–8], except for CL score that provided only aggregated data. CL score original distribution was 33.07% score 0–1, 37.27% score 1.5–2.5, 25.55% score 3.5–4.5 and 4.11% score ≥5 compared with the Iberian Registry 25.6%, 36.2%, 30.5% and 7.7% in the

**Table 2. Clinical characteristics.**

| Variable | |
|---|---|
| Age | 65.17 ± 11.11 |
| Sex (male) | 84.7% |
| Hypertension | 68.26% |
| Dyslipidemia | 67.56% |
| Diabetes | 35.39% |
| Smoker | 42.70% |
| Creatinin (mg/dl) | 1.16 ± 3.77 |
| Chronic kidney disease | 11.67% |
| Previous CABG | 6.79% |
| Previous PCI | 49.18% |
| Previous MI | 29.62% |
| Previous stroke | 5.70% |
| Peripheral vascular disease | 10.26% |
| Multivessel coronary disease | 58.05% |
| Syntax Score | 17.3 ± 12.58 |
| LVEF (%) | 49.42 ± 17.54 |
| CTO location | |
| Left main | 0.22% |
| Left anterior descending | 32.56% |
| **Left circumflex** | 16.84% |
| Right coronary artery | 50.37% |

**Table 3. PCI procedure characteristics.**

| Variable | |
|---|---|
| Redo attempt | 13.93% |
| Primary CTO approach | |
| Antegrade | 80.5% |
| Retrograde | 6.6% |
| Hybrid | 11% |
| Unknown | 1.9% |
| Successful technique for wire crossing# | |
| Wire escalation | 54.7% |
| Parallel wire / see saw | 5.5% |
| CART or reverse CART | 4.8% |
| Balloon-assisted reentry | 2% |
| Not disclosed# | 33% |
| Intravascular Ultrasound use | 14% |
| Drug eluting stent (vs. bare metal) | 99.1% |
| Total stent length | 41.8 ± 33.1 |
| Stent diameter | 2.8 ± 0.44 |
| Periprocedural complications | 5.20% |
| Cardiac tamponade | 0.89% |
| Myocardial infarction | 0.75% |
| Perforation without tamponade | 0.67% |
| Vascular access | 0.52% |
| Heart Failure | 0.37% |
| Coronary dissection (remote to CTO segment) | 0.37% |
| Death | 0.22% |
| Septal hematoma | 0.15% |
| Life-threatening arrhythmia | 0.15% |
| Others | 1.12% |
| Contrast (ml) | 254 ± 175.74 |
| Fluoroscopy time (min) | 38.19 ± 37.31 |
| Radiation (mGy) | 2419 ± 2107.91 |

Data are percentages or mean±standard deviation. CABG, Coronary Artery Bypass Graft. CTO, Chronic Total Occlusion. LVEF, Left Ventricular Ejection Fraction. MI, Myocardial Infarction. PCI, Percutaneous Coronary Intervention. mGy, MiliGrays. # This was an elective variable in the database

same categories. The distributions show that the Iberian Registry cohort has lower complexity than CASTLE derivation cohort but higher than J CTO, PROGRESS or CL derivation cohorts. Fig 2 shows the predicted and observed success rates per each possible score's category (obtained by logistic regression analysis). All scores showed a trend towards an inverse linear relationship between procedural success rate with score values ($p < 0.001$ in all scores), but in the upper range of predicted scores the actual success rate was higher than expected, irrespective of the employed score. On the contrary, lower score values overestimated the actual success rate. The CASTLE and CL scores were well-calibrated using the Hosmer-Lemeshow goodness-of-fit test ($p > 0.05$), but not the PROGRESS and J-CTO scores ($p < 0.05$). Note that although in the highest complexity strata of CASTLE and CL scores (5–6 and 7–8 points respectively) the success rates were higher than expected, they represent <1% of the study population (see Fig 1).

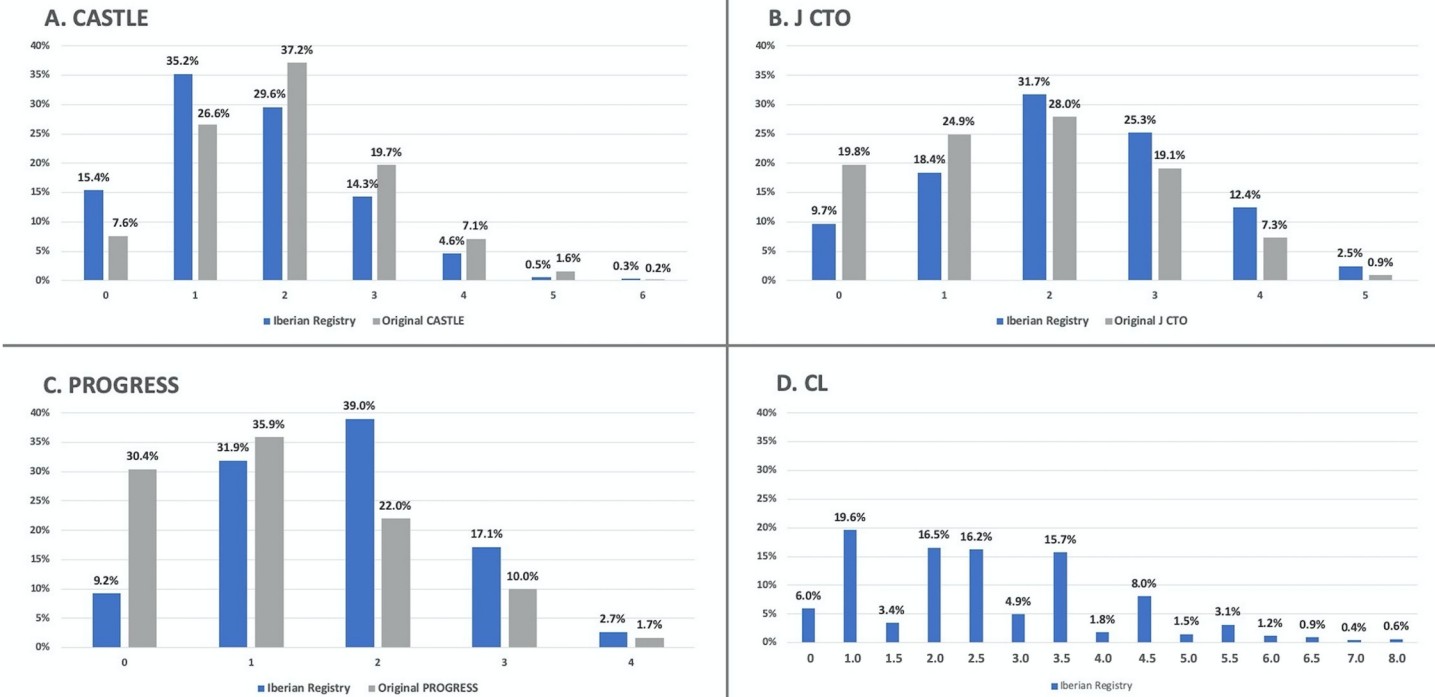

**Fig 1. Barchart of scoring distribution.** The study population scoring distribution (n = 1342 CTOs) is shown in blue. Original derivation cohort scoring distribution is shown in gray, but for CL score (not available in original publication). Note that 0.5 and 7.5 are not possible to be obtained in CL-score.

The discrimination of the scores for procedural success was tested using the AUC of the ROC curve (Fig 3). AUC from PROGRESS was significantly lower than CASTLE AUC, but J-CTO and CL score AUC were not different from CASTLE AUC. However, the overall discriminatory capacity of all scores was limited (as a consensus AUC <0.7 is considered poor to moderate discrimination [13]). Comparing CASTLE to J-CTO, we found no differences in reclassification abilities as evaluated with IDI /NRI (Table 4); however, CASTLE was superior to PROGRESS (IDI +0.036, p<0.001 and NRI +0,379, p<0.001). Finally, CASTLE had slightly inferior IDI than CL score (– 0.011, p = 0.004), but similar NRI (p = 0.31).

Table 5. shows sensitivity, specificity, positive predictive value (PPV), negative predictive value (NPV) and Youden index values at "complex CTO" cutoffs (Table 5A). We also estimated these values at best Youden index value cutoffs (Table 5B) which were CASTLE <2, J-CTO <3, PROGRESS <2, CL ≥2.5.

## Discussion

This study provides a comprehensive comparison of the new CASTLE score against the most commonly used CTO scores in an extensive, national database of CTO procedures, including expert and learning CTO operators. However, we found poor-to-intermediate performance in the c-statistic predicting CTO success among all scores. With small differences, CASTLE score performed best along with CL score, followed by J-CTO and PROGRESS with slightly worse efficiency. An interesting lesson of this study is that applying different scores to the same cohort, the spectrum of difficulty is variable and not the same as in the original CTO scores cohorts (Fig 1). In other words, there seems to be a lack of consistency among them. How, then, a CTO score should be chosen?

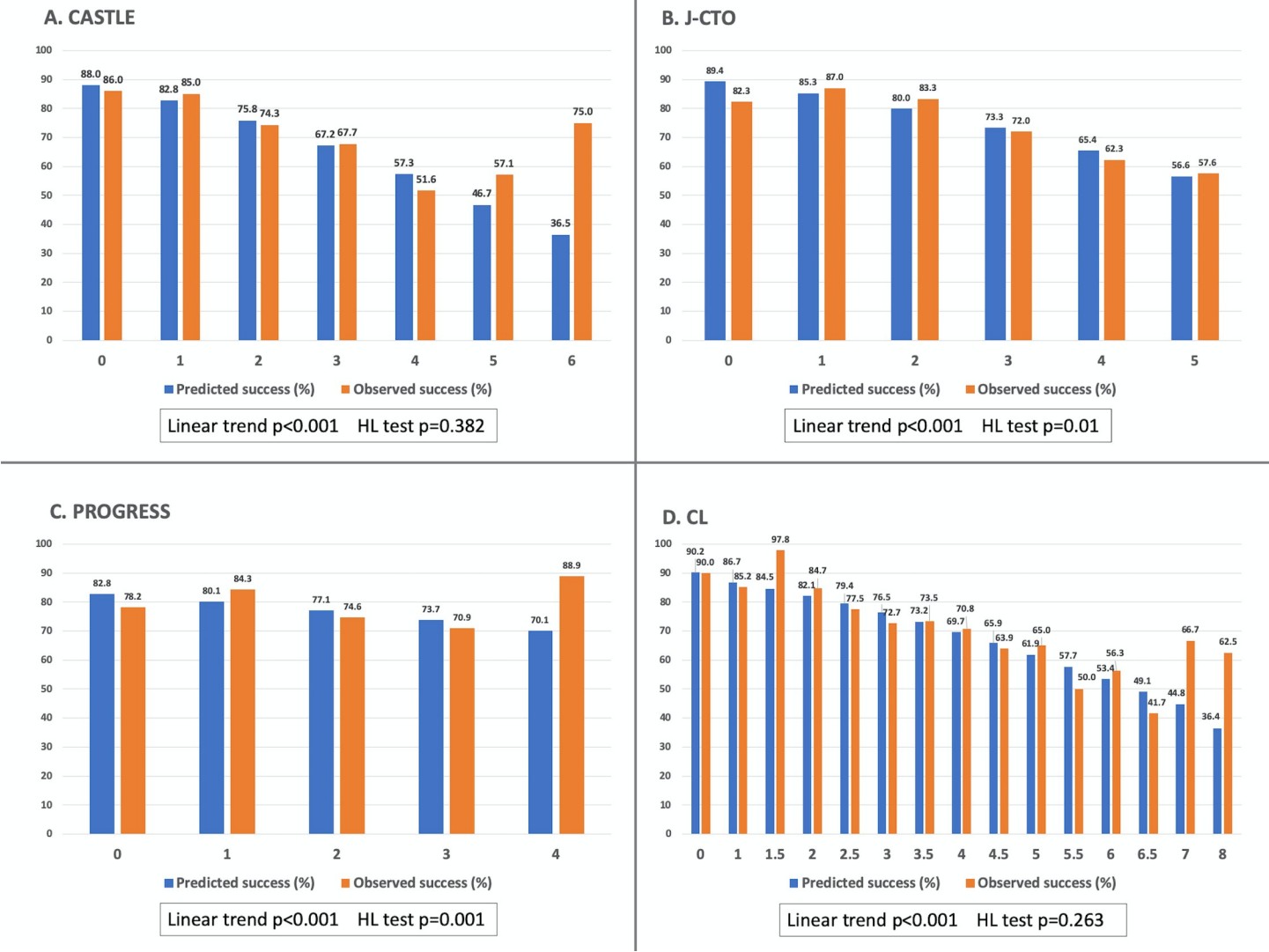

**Fig 2. Expected success rates versus observed success rates across different strata of each score.** P values for linear trend and Hosmer-Lemeshow (HL) tests are provided.

The probability of success in CTO PCI is dependent on multiple factors. On the one hand, on the operator's experience and the availability and use of an extensive toolbox. On the other, it relies on the use of a structured approach, in which the use of CTO PCI scores plays an essential role [2, 14]. First, they allow the operator to gauge the feasibility of procedural success according to his/her level of expertise, particularly over the learning curve of CTO PCI. Second, they facilitate Heart Team discussions in cases in which CTO lesions are critical targets in an achieving equivalent degree of myocardial revascularization. And third, they provide insights on the procedural time, amount of contrast, radiation dose and risk of complications associated with the intervention that can be integrated into the decision-making process and in PCI planning [14].

CASTLE is the CTO score derived from the largest dataset (14,882 patients taken from 2008 to 2014), encompassing a broader number of operators, techniques and practices across Europe. Some relevant differences compared to previous scores must be pointed out. The

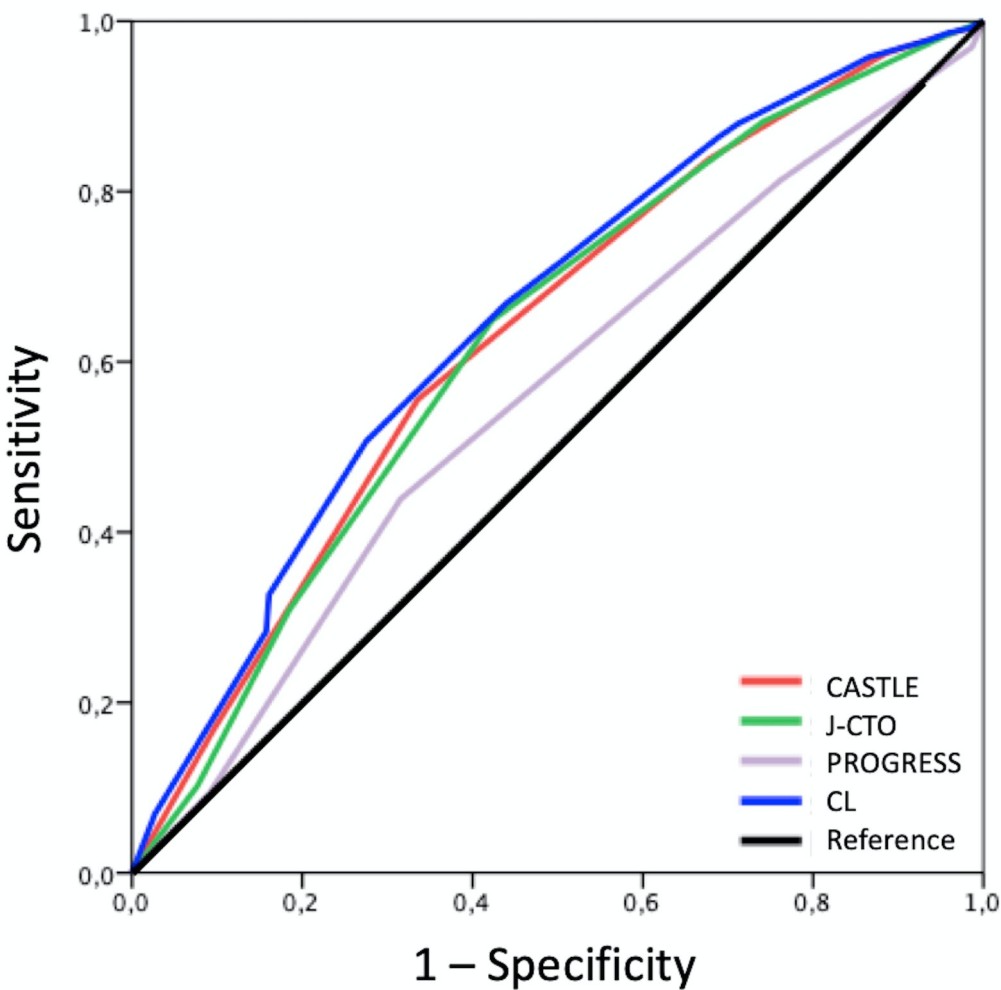

**Fig 3. ROC curve and AUC of each score discriminating procedural success.** Comparison between AUCs were done taking CASTLE score as a reference.

J-CTO was derived from a multi-centre Japanese database comprising 400 procedures (2006–2007), and designed to estimate the likelihood of passing an antegrade guidewire in less than 30 minutes [5]. Posteriorly, it was extensively validated as success and even outcomes predictor [15, 16]. The CL score was designed to assess procedural failure in a first CTO-PCI procedure,

**Table 4. Integrated Discrimination Improvement Index (IDI) and Net Reclassification Improvement (NRI) of JCTO, PROGRESS and CL scores compared to CASTLE score as a reference.**

|  | IDI | Continuous NRI (overall) | Continuous NRI event (%) | Continuous NRI nonevent (%) |
|---|---|---|---|---|
| **JCTO** | 0.00496 (p = 0.335) | -0.00296 (p = 0.964) | 8.43 | -8.72 |
| **PROGRESS** | 0.03636 (p<0.001) | 0.379 (p<0.001) | 17.82 | 20.13 |
| **CL** | -0.01121 (p = 0.004) | -0.0662 (p = 0.313) | 12.84 | -19.46 |

including for the first-time clinical variables. However, it was derived from a single-centre European cohort of mainly (90.7%) antegrade CTO cases (n = 1,671, from 2004 to 2013) [6]. PROGRESS CTO was derived from a multi-centre US database, is more contemporary to CASTLE (2012–2015), and designed to assess technical success using the hybrid approach [7]. These three were chosen as comparators because they are the most commonly used scores and represent different regional approaches.

The heterogeneous derivation cohorts probably explain that the different scores include a heterogeneous set of variables, except for the blunt proximal cap (Table 1). Two scores include clinical variables. They concur in CABG, which is acknowledged as an adverse feature in CTO patients [17, 18]. Moreover, a recent study from the PROGRESS database found 5% less recanalization success in CABG patients compared to non-CABG patients [19]. CTO characteristics such as tortuosity and calcification are more challenging to be consistently evaluated, as they are defined discordantly among scores and might be somewhat subjective to the operator. However, we agree that differently to the softer J-CTO had definitions [5], severe tortuosity and severe calcification are common ground, especially in combination, for challenging CTO scenarios. Finally, it is remarkable that only one score (PROGRESS) includes assessing collaterals in the scoring, which is very relevant for CTO planning. Despite these differences, we genuinely believe that the careful CTO evaluation needed to calculate one or more scores is valuable for procedural planning, especially for the less experienced operator.

Comparing our cohort with the derivation cohorts of previous scores we see a shift towards higher complexity in the REBECO Registry compared to older J CTO, CL and PROGRESS scores (Our cohort starts at year 2015 while J CTO, CL and PROGRESS have older derivation cohorts [5–7]). This is probably as a result of widespread standardization of techniques and equipment allowing CTO operators to tackle more complex cases [2]. In contrast, the pragmatic nature of the Iberian Registry includes centres and operators at different points of the CTO learning curve, quite differently from, for example, the CASTLE data derived from the highly expert EuroCTO Club [8]. Consequently (and saving the differences in endpoints), we

**Table 5. Sensitivity, specificity, positive predictive value (PPV), negative predictive value (NPV) and Youden index values at different cutoffs.**

| Score | Cutoff value | Sensitivity | Specificity | PPV | NPV | Youden index |
|---|---|---|---|---|---|---|
| **A. For binary (simple vs complex) CTO cutoffs** | | | | | | |
| **CASTLE** | <4 | 96.26% | 11.41% | 79.20% | 46.58% | 0.08 |
| **J-CTO** | <3 | 64.85% | 57.72% | 84.31% | 31.91% | 0.23 |
| **PROGRESS** | <3 | 81.32% | 23.83% | 78.90% | 26.69% | 0.05 |
| **CL** | <5.5 | 95.79% | 13.42% | 79.49% | 47.62% | 0.09 |
| **B. For maximal Youden index values cutoffs** | | | | | | |
| **CASTLE** | <2 | 55.56% | 66.44% | 85.29% | 29.91% | 0.22 |
| **J-CTO** | <3 | 64.85% | 57.72% | 84.31% | 31.91% | 0.23 |
| **PROGRESS** | <2 | 43.87% | 68.46% | 82.97% | 25.82% | 0.12 |
| **CL** | <2.5 | 50.67% | 72.48% | 86.58% | 29.55% | 0.23 |

had less recanalization success (77.8%) compared to 84.2% in CASTLE, 92.5% in PROGRESS and 88.6% in J CTO; but better than the CL cohort (72.5%) [5–8].

Many of the newly developed scores compared themselves in the original reports to J-CTO, showing better parameters of calibration and discrimination [5–8]. However, this calculation might be biased because the validation and derivation cohort come together from a single "mother" cohort that is likely to perform better with its derived score than with an external score (J-CTO). Some score comparisons have been previously published showing that the performance of the scores might be similar. Karastakis et al. compared CL, J-CTO and PROGRESS scores in a cohort from the PROGRESS CTO registry (n = 664), showing similarly poor to moderate (<0.7) discrimination as evaluated per AUC (with no inter-score differences) [20]. This analysis might be biased because the study sample also comes from the PROGRESS CTO Registry. Recently, Kalogeropoulos et al. used an international database (n = 660) to compare CASTLE to J-CTO, finding equal overall discriminatory capacity of both scores (AUC 0.676 and 0.698 respectively). CASTLE outperformed J-CTO in the most complex cases (J-CTO $\geq$3 or CASTLE $\geq$4 representing only roughly 9% of the sample) but with quite low overall AUC (0.588). Our comparison study has the strengths of a more extensive, independent testing database (n = 1,342) and a more comprehensive analysis using four scores and taking CASTLE score as a reference.

In our study, the calibration (meaning how close the observed and expected results were) was better for CASTLE and CL scores, although this test does not allow for inter-score comparisons. However, the differences are rather small (Fig 2). The discrimination measured with AUC and with the IDI index was better for CASTLE, J-CTO and CL and slightly worse for PROGRESS score; however poor to moderate (<0.7) in absolute terms, in agreement with previous publications [10, 20]. Complementary to the AUC that has some limitations [21], we assessed the incremental value of the newer CASTLE score using two reclassification indexes, the Integrated Discrimination Improvement (IDI) and the Net Reclassification Improvement (NRI) indices [12, 22]. The IDI, which is possibly more sensitive than the AUC comparison showed that CASTLE had better discrimination than PROGRESS, similar to J-CTO and slightly inferior than CL. The NRI analysis showed that CASTLE reclassified cases better than PROGRESS in roughly 20% of cases (CASTLE correctly reclassified 17.82% of event cases and 20.13% of nonevent cases into a higher or lower predicted risk of success, respectively). However, CASTLE did not significantly improve reclassification compared to J-CTO and CL scores. The intermediate or poor performance of current available CTO scores in predicting CTO success suggests the need for more precise mechanisms to predict the outcomes and precisely inform our patients.

We provided in Table 4 data on sensitivity, specificity, PPV, NPV and Youden indexes showing only modest values with commonly used complex CTO cutoffs and cutoffs that maximizes the Youden index. However, we believe that binary categorization is unnecessary in CTO scores because it masks the spectrum of complexity provided by scores. Furthermore, PPV and NPV provide insufficient precision to inform for or against attempting a specific CTO case (highly expert operators demonstrated high J-CTO scores being non-associated with observed success rates [23]).

The information provided by our study may also help in selecting a specific CTO PCI score for particular purposes. Very experienced operators with success rates over 90% will take little interest on the success discrimination capabilities but might use CASTLE or CL in order to discuss efficiency and complication risks with patients (both have the advantage of combining angiographic and clinical variables; CASTLE is probably more intuitive to calculate and has fewer categories than CL score). Less experienced operators might choose CASTLE or CL scores on the grounds of better calibration and discrimination to predict which highly complex

cases might benefit for proctoring or referral (although we must bear in mind that in this or any other study the overall discrimination is poor to moderate). Also, the predicted success rates might be easily obtained with univariate logistic regression analysis in a local database, using one or more scores to choose the one with the best "personalized" calibration. For research and benchmarking, J-CTO is the oldest and most widespread score and thus is critical to allow comparison with earlier studies. CASTLE is without a doubt the score with more solid foundations in terms of contemporaneity and derivation dataset size, so it will be probably a reference for future publications.

A few limitations should be reported regarding this study. First, the original angiographies were not assessed by a central core lab; we trusted in the individual investigator's evaluation of each item contributing to the different scores. Second, the collected data is self-reported and not systematically audited, so in spite of quality control some degree of selection and reporting bias are possible. Third, although this is a multi-centre, contemporary database, it might not be representative of specific practices, strategies or skillsets. Finally, more scores might be considered for comparison, although as discussed before, we found these as the most commonly used scores.

Several available CTO scores were not assessed in our study on the grounds of the preferential use of a specific device or strategy (CrossBoss and hybrid techniques in Europe, RECHARGE registry [24]); derivation from a single operator's experience (ORA score [25]); or the need for interesting but non-mandatory methods in CTO assessment (CT-RECTOR [26] or KCCT [27] scores). However, many predictors of procedural failure are common: stump, calcification, tortuosity, length, previous CABG.

## Conclusion

We compared 4 CTO recanalization success scores in a large, independent, multicenter database. Overall discrimination was poor to moderate with c-statistics predicting CTO success below 0.7 among all scores. CASTLE score performed best along with CL score, followed by J-CTO and PROGRESS with slightly worse efficiency.

Procedural PCI difficulty is not consistently depicted by available CTO scores and is probably influenced by the characteristics of each CTO cohort. Operators in different points of their learning curve should be aware and consider the choice of an adequate score for a specific purpose. In the case of our study population, including expert and learning operators, the CASTLE score had slightly better overall performance along with CL score.

## Author Contributions

**Conceptualization:** Pablo Salinas, Victoria Martin-Yuste, Manel Sabaté, Beatriz Vaquerizo, Javier Escaned.

**Data curation:** Pablo Salinas, Víctor H. Moreno, Manuel Fuentes, Francisco Bosa Ojeda, Juan Caballero Borrego, Javier Cuesta, José María de la Torre Hernández, Alejandro Diego-Nieto, Daniela Dubois, Guillermo Galeote, Javier Goicolea, Alejandro Gutiérrez, Miriam Jiménez-Fernández, Jesús Jiménez-Mazuecos, Alfonso Jurado, Javier Lacunza, Dae-Hyun Lee, María López, Fernando Lozano, Javier Martin-Moreiras, Victoria Martin-Yuste, Raúl Millán, Gema Miñana, Mohsen Mohandes, Francisco J. Morales-Ponce, Julio Núñez, Soledad Ojeda, Manuel Pan, Fernando Rivero, Javier Robles, Sergio Rodríguez-Leiras, Sergio Rojas, Juan Rondán, Eva Rumiz, Manel Sabaté, Juan Sanchís, Beatriz Vaquerizo.

**Formal analysis:** Manuel Fuentes, Beatriz Vaquerizo.

**Funding acquisition:** Victoria Martin-Yuste.

**Investigation:** Nieves Gonzalo, Víctor H. Moreno, Sandra Santos-Martinez, José Antonio Fernandez-Diaz, Ignacio J. Amat-Santos, Francisco Bosa Ojeda, Juan Caballero Borrego, Javier Cuesta, José María de la Torre Hernández, Alejandro Diego-Nieto, Daniela Dubois, Guillermo Galeote, Javier Goicolea, Alejandro Gutiérrez, Miriam Jiménez-Fernández, Jesús Jiménez-Mazuecos, Alfonso Jurado, Javier Lacunza, Dae-Hyun Lee, María López, Fernando Lozano, Javier Martin-Moreiras, Victoria Martin-Yuste, Raúl Millán, Gema Miñana, Mohsen Mohandes, Francisco J. Morales-Ponce, Julio Núñez, Soledad Ojeda, Manuel Pan, Fernando Rivero, Javier Robles, Sergio Rodríguez-Leiras, Sergio Rojas, Juan Rondán, Eva Rumiz, Manel Sabaté, Juan Sanchís, Beatriz Vaquerizo.

**Methodology:** Pablo Salinas, Manuel Fuentes, Javier Escaned.

**Project administration:** Pablo Salinas, José Antonio Fernandez-Diaz, Ignacio J. Amat-Santos.

**Supervision:** Beatriz Vaquerizo, Javier Escaned.

**Writing – original draft:** Pablo Salinas.

**Writing – review & editing:** Pablo Salinas, Nieves Gonzalo, Víctor H. Moreno, Manuel Fuentes, Sandra Santos-Martinez, José Antonio Fernandez-Diaz, Ignacio J. Amat-Santos, Francisco Bosa Ojeda, Juan Caballero Borrego, Javier Cuesta, José María de la Torre Hernández, Alejandro Diego-Nieto, Daniela Dubois, Guillermo Galeote, Javier Goicolea, Alejandro Gutiérrez, Miriam Jiménez-Fernández, Jesús Jiménez-Mazuecos, Alfonso Jurado, Javier Lacunza, Dae-Hyun Lee, María López, Fernando Lozano, Javier Martin-Moreiras, Victoria Martin-Yuste, Raúl Millán, Gema Miñana, Mohsen Mohandes, Francisco J. Morales-Ponce, Julio Núñez, Soledad Ojeda, Manuel Pan, Fernando Rivero, Javier Robles, Sergio Rodríguez-Leiras, Sergio Rojas, Eva Rumiz, Manel Sabaté, Juan Sanchís, Beatriz Vaquerizo, Javier Escaned.

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
