## [Decision Letter · Decision Letter 0]

15 Dec 2020

PONE-D-20-30053

Choice of CTO scores to predict procedural success in clinical practice. A comparison of 4 different CTO PCI scores in a comprehensive national registry including expert and learning CTO operators.

PLOS ONE

Dear Dr. Salinas,

Thank you for submitting your manuscript to PLOS ONE. After careful consideration, we feel that it has merit but does not fully meet PLOS ONE’s publication criteria as it currently stands. Therefore, we invite you to submit a revised version of the manuscript that addresses the points raised during the review process.

Reviewers have raised concerns -among other issues- about the lack of data reporting the relative use of different techniques to achieve vessel recanalization in order to appreciate the actual complexity of the cases included and how to account for different definitions across the diverse scores. Reviewers have highlighted that the value of scores should be put in perspective according to the skills and experience of the operators. 

We look forward to receiving your revised manuscript.

Kind regards,

Giuseppe Andò, M.D., Ph.D.

Academic Editor

PLOS ONE

Journal Requirements:

2. One of the noted authors is a group or consortium [REBECO collaborators]. In addition to naming the author group, please list the individual authors and affiliations within this group in the acknowledgments section of your manuscript. Please also indicate clearly a lead author for this group along with a contact email address.

Reviewers' comments:

Reviewer's Responses to Questions

**Comments to the Author**

1. Is the manuscript technically sound, and do the data support the conclusions?

Reviewer #1: Yes

Reviewer #2: Yes

Reviewer #3: Yes

2. Has the statistical analysis been performed appropriately and rigorously? 

Reviewer #1: Yes

Reviewer #2: Yes

Reviewer #3: Yes

3. Have the authors made all data underlying the findings in their manuscript fully available?

Reviewer #1: Yes

Reviewer #2: Yes

Reviewer #3: Yes

4. Is the manuscript presented in an intelligible fashion and written in standard English?

Reviewer #1: Yes

Reviewer #2: Yes

Reviewer #3: Yes

5. Review Comments to the Author

Reviewer #1: Everybody knows the importance to calculate scores in order to quickly understand the complexity of the intervention, but we also know that complexity is proportional to the operators expertise and clinical comorbidities. For this reason it would be desirable to use the best score just for learning operators. This is a real world comparison between different scores in unselected patients underwent CTO recanalization by expert and learning CTO operators. It was interesting to read this manuscript. I have nothing to add. Well done.

Reviewer #2: In this paper the authors wanted to compare 4 CTO scores to predict the success in CTO PCI in the Spanish National Registry (REBECO). This regitry included expert and learning CTO operators, including more than 1300 patients treated in the period 2015-2019.

The 4 scores have different anatomical variables that have been integrated in order to perform an objective assesment of procedural CTO difficulty. The most recent score is the CASTLE score, developed from EuroCTO database, with more than 14.000 patients included. Other three scores were: J CTO score (Japan), Progress CTO score (USA), CL score (France)

The statistical evaluation in the paper considered the comparison between Spanish registry and the 4 scores in therms of calibration, discrimination and reclassification.

The statistical analysis was very well developed and conducted, resulting in a slightly better overal performance of CASTLE score , but with only a poor to intermediate performance in the prediction of CTO PCI success among all scores.

The authors concluded that the probability of success in these procedures depends on multiple factors, and the most important is operator's expertise.

I think that CTO scores are really important for many reasons especially for operators at the beginning of their learning curve, because in that group of operator the correlation of CTO score complexity and success rate can be important. One young operator with success rate near 70%, can have a great help in selectong cases that are not so complex in order to reduce the risk of complication and increase their success rate.

On the contrary in expert operators, with a success rate more than 90% overall, the use of a CTO score is less importan, because with the modern approach to CTO PCI, hybrid approach, one operator needs to be flexible during the procedure and he is able to change many techniques during the same procedure, also in complex cases with high CTO scores.

Moreover the use of CTO scores is helping the operator in the most important phase of the procedure, that means the pre.procedural phase of planning and careful evaluation of the anatomical features of the CTO lesion (lenght, calcium, tortuosity, prox cap ambiguity, presence of interventional collaterals).

Then less experienced operator can evaluate with CTO scores which complex patients might benefit for proctoring or referred to a more experienced operator.

The authors conclusion is appropriate with the data presented in the paper .

Reviewer #3: The authors used a national CTO database to compare the performance of 4 different CTO success prediction scores (CASTLE, J-CTO, PROGRESS and CL). They concluded that the CASTLE and CL scores had a slightly better performance overall, but the predicting ability of all scores was at most moderate.

Up-take of CTO PCI increases and therefore the authors’ attempt to identify the most accurate score, which would facilitate the multiple aspects of CTO PCI decision-making and procedural planning, is interesting and clinically relevant. The study is well conducted and the manuscript well written.

However, I have the following comments/queries:

- The authors do not report the successful mode of vessel recanalization (Antegrade Wire Escalation vs. Antegrade Dissection Re-entry vs. Retrograde approach). Therefore, the adoption of advanced contemporary CTO techniques in the described cohort is unknown. This information is important in order to appreciate the complexity of the cases, the skills of the operators involved and to put the described cohort in a contemporary and comparable setting. Furthermore, other essential procedural information (e.g. stent length and size) are missing.

- Although, the authors briefly discuss the derivation cohorts for the 4 scores, they do not discuss at all their different clinically relevant variables. For example, it is common knowledge among experienced CTO operators that CTO PCI in previous CABG cases (a scoring point for CASTLE and CL) is more challenging. On the other hand, how accurate can a score be when it does not include an assessment of collateral channels and the retrograde approach is the successful strategy in up to 1/3 of the cases in contemporary expert practice? The above discussion is an important part of any manuscript examining CTO PCI predictions scores.

- Tortuosity and calcification have different definitions in different scores. How was this problem addressed during score calculation since the angiograms were not re-assessed?

- The REBECO registry has a voluntary character and the data collected are self-reported. Is there a quality/validity assessment process for the collected data? Selection and reporting bias should be recognised as a limitation of the study.

- According to the authors the registry includes expert and learning operators. An analysis based on the level of experience would be interesting.

- The authors report periprocedural complications with a rate of 5.2%. Is there a breakdown for these complications?

6. PLOS authors have the option to publish the peer review history of their article (what does this mean?). If published, this will include your full peer review and any attached files.

Reviewer #1: No

Reviewer #2: **Yes: **Roberto Garbo, MD

Maria Pia Hospital

GVM Care & Research

Turin, Italy

Reviewer #3: **Yes: **Dr Grigoris Karamasis

---

## [Author Response · Author response to Decision Letter 0]

4 Jan 2021

Editorial comments

Comments to the Author

Reply: The R1 manuscript was thoroughly revised and adapted to PLOS ONE's style requirements, including those for file naming.

2) One of the noted authors is a group or consortium [REBECO collaborators]. In addition to naming the author group, please list the individual authors and affiliations within this group in the acknowledgments section of your manuscript. Please also indicate clearly a lead author for this group along with a contact email address.

Reply: Regarding the research consortium 'REBECO Group' to which we belong, we include the full list of investigators in the acknowledgements, instead of a supplementary file (therefore there are no supplementary files now in the article). We respectfully ask the editorial board to include the collaborator/investigator names that do not qualify as article's authors in the article by-name to be indexed in Medline as such (https://www.nlm.nih.gov/bsd/policy/authorship.html). Therefore, we included the investigator names on the first page (as well as in acknowledgements).

The leading investigator of REBECO group is Jose Antonio Fernández (joseantoniofer@gmail.com). This information is given in the acknowledgements.

3) Please include captions for your Supporting Information files at the end of your manuscript, and update any in-text citations to match accordingly. Please see our Supporting Information guidelines for more information: http://journals.plos.org/plosone/s/supporting-information.

Reply: As commented before there are no longer supporting Information files

Reviewer #1

1) Everybody knows the importance to calculate scores in order to quickly understand the complexity of the intervention, but we also know that complexity is proportional to the operators expertise and clinical comorbidities. For this reason it would be desirable to use the best score just for learning operators. This is a real world comparison between different scores in unselected patients underwent CTO recanalization by expert and learning CTO operators. It was interesting to read this manuscript. I have nothing to add. Well done.

Reply: Thank you for your kind words on our manuscript

Reviewer #2:

1) In this paper the authors wanted to compare 4 CTO scores to predict the success in CTO PCI in the Spanish National Registry (REBECO). This regitry included expert and learning CTO operators, including more than 1300 patients treated in the period 2015-2019.

The 4 scores have different anatomical variables that have been integrated in order to perform an objective assesment of procedural CTO difficulty. The most recent score is the CASTLE score, developed from EuroCTO database, with more than 14.000 patients included. Other three scores were: J CTO score (Japan), Progress CTO score (USA), CL score (France)

The statistical evaluation in the paper considered the comparison between Spanish registry and the 4 scores in therms of calibration, discrimination and reclassification.

The statistical analysis was very well developed and conducted, resulting in a slightly better overal performance of CASTLE score , but with only a poor to intermediate performance in the prediction of CTO PCI success among all scores. The authors concluded that the probability of success in these procedures depends on multiple factors, and the most important is operator's expertise.

I think that CTO scores are really important for many reasons especially for operators at the beginning of their learning curve, because in that group of operator the correlation of CTO score complexity and success rate can be important. One young operator with success rate near 70%, can have a great help in selectong cases that are not so complex in order to reduce the risk of complication and increase their success rate.

On the contrary in expert operators, with a success rate more than 90% overall, the use of a CTO score is less importan, because with the modern approach to CTO PCI, hybrid approach, one operator needs to be flexible during the procedure and he is able to change many techniques during the same procedure, also in complex cases with high CTO scores. Moreover the use of CTO scores is helping the operator in the most important phase of the procedure, that means the pre.procedural phase of planning and careful evaluation of the anatomical features of the CTO lesion (lenght, calcium, tortuosity, prox cap ambiguity, presence of interventional collaterals).

Then less experienced operator can evaluate with CTO scores which complex patients might benefit for proctoring or referred to a more experienced operator.

The authors conclusion is appropriate with the data presented in the paper . 

Reply: Thank you for your nice words on our study. We agree with your opinion, and we added some of your interesting thoughts in the discussion: 

Despite these differences, we genuinely believe that the careful CTO evaluation needed to calculate one or more scores is valuable for procedural planning, especially for the less experienced operator.

Reviewer #3

The authors used a national CTO database to compare the performance of 4 different CTO success prediction scores (CASTLE, J-CTO, PROGRESS and CL). They concluded that the CASTLE and CL scores had a slightly better performance overall, but the predicting ability of all scores was at most moderate. 

Up-take of CTO PCI increases and therefore the authors’ attempt to identify the most accurate score, which would facilitate the multiple aspects of CTO PCI decision-making and procedural planning, is interesting and clinically relevant. The study is well conducted and the manuscript well written. 

However, I have the following comments/queries: 

1) The authors do not report the successful mode of vessel recanalization (Antegrade Wire Escalation vs. Antegrade Dissection Re-entry vs. Retrograde approach). Therefore, the adoption of advanced contemporary CTO techniques in the described cohort is unknown. This information is important in order to appreciate the complexity of the cases, the skills of the operators involved and to put the described cohort in a contemporary and comparable setting. Furthermore, other essential procedural information (e.g. stent length and size) are missing. 

Reply: Thank you for your interest. We limited the amount of information on technical details to focus on the statistical analyses comparing the scores. We substituted the variable: ‘Primarily retrograde or hybrid approach’ in Table 2 for a more detailed description ‘Primary CTO approach’. We also added to Table 2 the following data: ‘Successful technique for wire crossing’, ‘Intravascular Ultrasound use’, ‘Drug eluting stent (vs bare metal)’, ‘Stent length’, ‘Stent diameter’. As Table 1 became too large, we decided to break it in two pieces (clinical characteristics and interventional characteristics)

Primary CTO approach

 Antegrade

 Retrograde

 Hybrid 

 Unknown 

80.5%

6.6%

11%

1.9%

Successful technique for wire crossing#

 Wire escalation

 Parallel wire / see saw

 CART or reverse CART

 Balloon-assisted reentry

 Not disclosed 

54.7%

5.5%

4.8%

2%

33%

Intravascular ultrasound use 14%

Drug eluting stent (vs bare metal) 99.1%

Stent length 41.8 �33.1

Stent diameter 2.8 � 0.44

2) Although, the authors briefly discuss the derivation cohorts for the 4 scores, they do not discuss at all their different clinically relevant variables. For example, it is common knowledge among experienced CTO operators that CTO PCI in previous CABG cases (a scoring point for CASTLE and CL) is more challenging. On the other hand, how accurate can a score be when it does not include an assessment of collateral channels and the retrograde approach is the successful strategy in up to 1/3 of the cases in contemporary expert practice? The above discussion is an important part of any manuscript examining CTO PCI predictions scores. 

Reply: We agree with the reviewer on the heterogeneity of the different variables included in the scores. We added a paragraph on this issue at Discussion: 

 The heterogeneous derivation cohorts probably explain that the different scores include a heterogeneous set of variables, except for the blunt proximal cap (Table 1). Two scores include clinical variables. They concur in CABG, which is acknowledged as an adverse feature in CTO patients(18,19). Moreover, a recent study from the PROGRESS database found 5% less recanalization success in CABG patients compared to non-CABG patients(20). CTO characteristics such as tortuosity and calcification are more challenging to be consistently evaluated, as they are defined discordantly among scores and might be somewhat subjective to the operator. However, we agree that differently to the softer J-CTO had definitions(5), severe tortuosity and severe calcification are common ground, especially in combination, for challenging CTO scenarios. Finally, it is remarkable that only one score (PROGRESS) includes assessing collaterals in the scoring, which is very relevant for CTO planning. Despite these differences, we genuinely believe that the careful CTO evaluation needed to calculate one or more scores is valuable for procedural planning, especially for the less experienced operator.

3) Tortuosity and calcification have different definitions in different scores. How was this problem addressed during score calculation since the angiograms were not re-assessed? 

Reply: This is an excellent question. As commented in the previous new paragraph, this might be a source of possible discrepancies in CTO features evaluation. Moreover, from the research standpoint, it’s very complex to host every different definition of each variable in a database. The CL score does not explicitly define the calcification (they only describe it as mild, moderate, or severe). We indeed did not re-review the angiograms (which was stated as a limitation), but we had what we believed was a good enough categorization of these variables in the REBECO database. We re-calculated the scores from the raw data using the following scheme:

Tortuosity in REBECO raw database CASTLE J CTO PROGRESS CL

No / Mild (<45º) 0 0 0 -

Moderate (≥ 45º <90º) 0 1 0 -

Severe (>90) 1 1 1 -

Calcification in REBECO raw database 

No 0 0 - 0

Mild (only during cine) 0 1 - 0

Moderate (visible in fluoro <50% CTO) 0 1 - 0

Severe (visible in fluoro ≥50% CTO) 1 1 - 1

4) The REBECO registry has a voluntary character and the data collected are self-reported. Is there a quality/validity assessment process for the collected data? Selection and reporting bias should be recognised as a limitation of the study. 

Reply: Yes, there is a risk of selection and reporting bias, as there is no monitoring of contributing centers and the data is self-reported. However, the Registry has the following quality controls:

1. The leading investigators of each center must be affiliated to the Spanish Interventional Cardiology Association (ACI-SEC), and therefore reports the yearly activity data to the general ACI-SEC Registry (administrative Registry with a limited set of variables such as the total number of different procedures, success rate, access, etc.), including the number of CTO procedures. The study coordinators compare the number of CTO procedures sent to the general ACI-SEC Registry and the REBECO Registry and ask the REBECO centers to reconcile the numbers in the REBECO database if there are discrepancies. If the center persists without including every CTO case, it is considered inactive.

2. The Registry is free to any center performing CTO to participate (therefore the mix of expert and learning CTO centers and the absolute number of CTO success). Still, the investigators are instructed to include consecutive cases. The Registry is open and enrolling patients, so if a center has a drop or a stop in CTO cases reporting a query is sent to this center. If the center discontinues the reporting to the database, it is deemed inactive and eventually closed for the Registry. 

We added in limitations the following sentence: 

Second, the collected data is self-reported and not systematically audited, so despite quality control, some degree of selection and reporting bias are possible.

5) According to the authors the registry includes expert and learning operators. An analysis based on the level of experience would be interesting. 

Reply: Thank you for your interest in this matter. In the first Registry report with the first 1000 patients the success rate varied with the center’s experience (Amat-Santos IJ, Martin-Yuste V, Fernández-Díaz JA, Martin-Moreiras J, Caballero-Borrego J, Salinas P, et al. Procedural, Functional and Prognostic Outcomes Following Recanalization of Coronary Chronic Total Occlusions. Results of the Iberian Registry. Rev Esp Cardiol Engl Ed. 1 de mayo de 2019;72(5):373-82.):

In the present study, we selected those cases with valid values on all critical variables used to estimate the four scores plus the variable CTO technical success (n=1342 CTO cases). As suggested, we performed a sensitivity analysis based on the volume of cases performed by center (quartiles of CTO volume), finding similar results with non-significant differences: 

 We also compared AUC for CTO success comparing centers above or below the p50, and above or below the p75 of procedures, finding no relevant differences:

 AUC full sample

(reported) AUC <p50 centers AUC >p50 centers AUC <p75 centers AUC >p75 centers

CASTLE 0.633 0.638 0.633 0.629 0.635

J-CTO 0.628 0.702 0.620 0.661 0.614

PROGRESS 0.557 0.601 0.552 0.567 0.554

CL 0.652 0.632 0.655 0.653 0.652

Therefore, we believe these sub-studies probably do not add value to the article, which is considerably long so far. However, we remain open at the reviewer's judgment on the possibility of adding these analyses as supplementary material.

6) The authors report periprocedural complications with a rate of 5.2%. Is there a breakdown for these complications? 

Periprocedural complications

 Cardiac tamponade

 Myocardial infarction

 Perforation without tamponade

 Vascular access

 Heart Failure

 Coronary dissection (remote to CTO segment)

 Death

 Septal hematoma

 Life-threatening arrhythmia

 Others 5.20%

0.89%

0.75%

0.67%

0.52%

0.37%

0.37%

0.22%

0.15%

0.15%

1.12%

Reply: We added the breakdown of these complications. As the Table 1 became too large, we decided to break it in two pieces (clinical characteristics and interventional characteristics).

---

## [Decision Letter · Decision Letter 1]

11 Jan 2021

Choice of CTO scores to predict procedural success in clinical practice. A comparison of 4 different CTO PCI scores in a comprehensive national registry including expert and learning CTO operators.

PONE-D-20-30053R1

Dear Dr. Salinas,

We’re pleased to inform you that your manuscript has been judged scientifically suitable for publication and will be formally accepted for publication once it meets all outstanding technical requirements.

Kind regards,

Giuseppe Andò, M.D., Ph.D.

Academic Editor

PLOS ONE

Additional Editor Comments (optional):

Reviewers' comments:

Reviewer's Responses to Questions

**Comments to the Author**

1. If the authors have adequately addressed your comments raised in a previous round of review and you feel that this manuscript is now acceptable for publication, you may indicate that here to bypass the “Comments to the Author” section, enter your conflict of interest statement in the “Confidential to Editor” section, and submit your "Accept" recommendation.

Reviewer #3: All comments have been addressed

2. Is the manuscript technically sound, and do the data support the conclusions?

Reviewer #3: Yes

3. Has the statistical analysis been performed appropriately and rigorously? 

Reviewer #3: Yes

4. Have the authors made all data underlying the findings in their manuscript fully available?

Reviewer #3: Yes

5. Is the manuscript presented in an intelligible fashion and written in standard English?

Reviewer #3: Yes

6. Review Comments to the Author

Reviewer #3: (No Response)

7. PLOS authors have the option to publish the peer review history of their article (what does this mean?). If published, this will include your full peer review and any attached files.

Reviewer #3: **Yes: **Grigoris Karamasis

---

## [Editor Report · Acceptance letter]

24 Mar 2021

PONE-D-20-30053R1 

Choice of CTO scores to predict procedural success in clinical practice. A comparison of 4 different CTO PCI scores in a comprehensive national registry including expert and learning CTO operators. 

Dear Dr. Salinas:

I'm pleased to inform you that your manuscript has been deemed suitable for publication in PLOS ONE. Congratulations! Your manuscript is now with our production department. 

Kind regards, 

on behalf of

Dr. Giuseppe Andò 

Academic Editor

PLOS ONE